# Bioproducts from *Passiflora cincinnata* Seeds: The Brazilian Caatinga Passion Fruit

**DOI:** 10.3390/foods12132525

**Published:** 2023-06-29

**Authors:** Carolina Cruzeiro Reis, Suely Pereira Freitas, Carolline Margot Albanez Lorentino, Thayssa da Silva Ferreira Fagundes, Virgínia Martins da Matta, André Luis Souza dos Santos, Davyson de Lima Moreira, Claudete Norie Kunigami, Eliane Przytyk Jung, Leilson de Oliveira Ribeiro

**Affiliations:** 1Laboratory of Vegetable Oil, Federal University of Rio de Janeiro, Rio de Janeiro 21941-909, Brazil; carolinacruzeiror@gmail.com (C.C.R.); freitasp@eq.ufrj.br (S.P.F.); 2Laboratory for Advanced Studies of Emerging and Resistant Microorganisms, Federal University of Rio de Janeiro, Rio de Janeiro 21941-902, Brazil; carolline.margot@gmail.com (C.M.A.L.); andre@micro.ufrj.br (A.L.S.d.S.); 3Laboratory of Natural Products, Rio de Janeiro Botanical Garden Research Institute, Rio de Janeiro 22460-030, Brazil; thayssafagundes@id.uff.br; 4Food Engineering Department, Embrapa Agroindústria de Alimentos, Rio de Janeiro 23020-470, Brazil; virginia.matta@embrapa.br; 5Post-Graduation Program in Translational Drugs and Medicines, Institute of Technology in Medicines, Oswaldo Cruz Foundation, Rio de Janeiro 21040-900, Brazil; 6Laboratory of Organic and Inorganic Chemical Analysis, National Institute of Technology, Rio de Janeiro 20081-312, Brazil; claudete.kunigami@int.gov.br (C.N.K.); eliane.jung@int.gov.br (E.P.J.)

**Keywords:** vegetable oil, antioxidant extract, spray dryer, microparticles, antibacterial activity

## Abstract

The present work aimed to obtain bioproducts from *Passiflora cincinnata* seeds, the Brazilian Caatinga passion fruit, as well as to determine their physical, chemical and biological properties. The seeds were pressed in a continuous press to obtain the oil, which showed an oxidative stability of 5.37 h and a fatty profile rich in linoleic acid. The defatted seeds were evaluated for the recovery of antioxidant compounds by a central rotation experimental design, varying temperature (32–74 °C), ethanol (13–97%) and solid–liquid ratio (1:10–1:60 *m*/*v*). The best operational condition (74 °C, 58% ethanol, 1:48) yielded an extract composed mainly of lignans, which showed antioxidant capacity and antimicrobial activity against Gram-positive and Gram-negative bacteria. The microencapsulation of linoleic acid-rich oil through spray drying has proven to be an effective method for protecting the oil. Furthermore, the addition of the antioxidant extract to the formulation increased the oxidative stability of the product to 30% (6.97 h), compared to microencapsulated oil without the addition of the antioxidant extract (5.27 h). The microparticles also exhibited favorable technological characteristics, such as low hygroscopicity and high water solubility. Thus, it was possible to obtain three bioproducts from the Brazilian Caatinga passion fruit seeds: the oil rich in linoleic acid (an essential fatty acid), antioxidant extract from the defatted seeds and the oil microparticles added from the antioxidant extract.

## 1. Introduction

The passion fruit is native to Latin America. The most used commercial species are the yellow passion fruit (*Passiflora edulis* Curtis) and the purple passion fruit (*Passiflora edulis* Sims). However, although less commercially explored, species such as *P. alata*, *P. quadrangularis* L., *P. cincinnata* Mast., and *P. mollisima* Bailey present unique sensory, chemical, and technological characteristics, increasing the fruit’s potential for edible and cosmetic purposes. Passion fruit juice and pulp are the main products derived from its processing. They can be used as ingredient in ice cream, jams, or even consumed *in natura* [1,2,3]. According to the Brazilian Institute of Geography and Statistics (IBGE), the global production of passion fruit was estimated to be 1.5 million tons, with Brazil being the largest producer, having produced about 1 million tons in 2021 [4].

*Passiflora cincinnata* is a native species to Brazil, cultivated in the North, Northeast and Southeast regions of the country. It is also found in other countries such as Argentina, Bolivia, Colombia, Paraguay, and Venezuela. This species is known as “maracujá-do-mato” or “maracujá-da-caatinga” and it has been mainly exploited by rural families’cooperatives in Bahia State (Northeast) in an extractive way. Its fruit is composed of 34% pulp, 26% seed and 40% peel [5]. Thus, its processing can generate 66% residues. As *Passiflora cincinnata* exhibits higher tolerance to water stress and pests when compared to other *Passiflora* species, this has motivated the Brazilian Agricultural Research Corporation, Embrapa, to develop cultivars with higher productivity and yield, and which contribute to the strengthening of the passion fruit agro-industrial chain in Brazil [6]. Thus, an increase in the generation of waste is envisioned, requiring studies that evaluate its potential in obtaining bioproducts with higher added value. Herein, it is important to highlight that studies with the residue of this species are still scarce in the literature, being restricted to the extraction of oil from seeds.

The use of passion fruit seeds to obtain oil has been the subject of much research since it represents a good way to add value to the agro chain of this fruit, as reported by Reis et al. [7]. Authors obtained oil of *Passiflora setacea*, *Passiflora alata*, and *Passiflora tenuifila* with increased oxidative stability by adding a hydroethanolic extract from fruit seeds. However, an optimization study to recover antioxidant compounds from defatted seeds has not been performed. Furthermore, both the composition of the extracts and the fatty acid profile of the oils have not been evaluated.

Products based on passion fruit seed oil are commercialized with functional appeal, which enhances its use and justifies the improvement of the extraction process for different *Passiflora* species and the application of alternative technologies to retain the oxidative stability of the oil and expand its use in the food and cosmetics industries [2,7].

Passion fruit seed oil contains polyunsaturated fatty acids, which are of interest to the pharmaceutical, cosmetic, and food industries. Moreover, bioactive compounds, mainly phenolics with antioxidant and anti-inflammatory properties, have been identified in oils of various *Passiflora* species [8,9]. Passion fruit seed oil contains 60–70% unsaturated fatty acids, consisting mainly of linoleic acid (C18:2), an essential fatty acid involved in cellular functions and in the formation of other acids in human metabolism [10,11]. In this way, it has a high nutritional value. On the other hand, linoleic acid is very susceptible to oxidation due to the low oxidative stability of polyunsaturated fatty acids.

Microencapsulation by spray drying is an already consolidated technique in food preservation and has been recently evaluated in the processing of edible oils. Due to its flexibility, using a spray dryer for microencapsulation of oils shows advantages when operating on a large scale. Additionally, the wall material used for emulsion preparation protects the encapsulated oil from light, oxygen, and moisture, for example [12,13]. In their previous study, Reis et al. [7] obtained antioxidant emulsions consisting of a mixture of oils from different passion fruit seeds (*Passiflora setacea*, *Passiflora alata*, and *Passiflora tenuifila*) and antioxidant extracts from their defatted seeds. Although these emulsions have shown higher oxidative stability when compared to pure oil, they are still very susceptible to oxidation. Thus, the use of microencapsulation can guarantee the chemical stability of both oil and antioxidant compounds. In addition to extending the shelf-life of the obtained products, the elaborated powders take up less space during storage when compared to liquid products such as emulsions and extracts, thus being a current approach to obtain bioproducts from *Passiflora cincinnata* seeds [13]. To our knowledge, this approach has not been evaluated for the biorefinery of seeds of this passion fruit species.

Therefore, the present work aimed to obtain bioproducts from passion fruit seeds (*P. cincinnata*) by lipid extraction, recovery of antioxidant compounds from defatted seeds and microencapsulation of seeds oil with antioxidant extract, as well as to determine the physical, chemical and biological properties of the obtained bioproducts.

## 2. Materials and Methods

### 2.1. Material

The seeds of *P. cincinnata* used in this work were obtained from fruits grown in experimental fields of Embrapa Semiarid (Petrolina, Brazil) after depulping. Seeds were stored at −20 °C until their processing. After thawing, the seeds were washed in a sieve under running water to remove the arils. Then, they were autoclaved at 120 °C for 20 min to reduce the microbial load.

### 2.2. Seeds Oil Extraction

The seeds were dried to reach about 10% moisture. After that, the raw material was ground in a knife mill (MA-048, Marconi, Piracicaba, SP, Brazil). Crushed passion fruit seeds were processed in continuous press (CA59G, Oekotec, Mönchengladbach, NRW, Germany) to obtain the first-extraction crude oil decanted for 24 h in a dark cabinet. The oil was centrifuged at 3000 rpm (CR22N, Hitachi Koki, Minato, Tokyo, Japan) for 15 min to separate very fine particles. Then, the clarified seeds oil was stored at −20 °C until use, and the pressed cake (defatted seeds) was reserved for antioxidant compound extraction.

The oil extraction yield was calculated from the ratio between the mass of oil recovered by cold pressing (m) and the mass of oil extracted by Soxhlet (M).
(1)$$Y\;\left(\%\right)=\frac{m}{M}*100$$

#### 2.2.1. Fatty Acid Composition

The fatty acid composition of *P. cincinnata* seeds oil samples was analyzed using gas chromatograph–mass spectrometry (GC–MS) (6890-5975, Agilent, Santa Clara, CA, USA). Fatty acid methyl esters were prepared and analyzed in accordance with Jung et al. [14], and the samples were injected into a Carbowax DB23 (60 m × 0.25 mm i.d. × 250 µm film) column; helium as carrier gas flowed at 1 mL min^−1^. The inlet temperature was 230 °C, the injection volume was 1 µL with a split ratio of 50:1, and the oven temperature was 200 °C for 40 min (isothermal). The transfer line temperature was 250 °C. The detected compounds were identified by matching their mass spectra with the reference spectra in the Willey7Nist05 library. The results were expressed in percentages based on the area of the chromatogram peaks without correction.

#### 2.2.2. Oxidative Stability

The oxidative stability of *P. cincinnata* seeds oil was measured using the Rancimat^®^ 743 (743, Metrohm, Riverview, FL, USA) equipment, passing a stream of purified air of 10 L h^−1^ at 110 °C through the oil sample (3 g). Results were expressed as an induction period representing a time interval until the sample reaches a high oxidation level.

### 2.3. Antioxidant Compound Recovery from Defatted Seeds

The *P. cincinnata* defatted seeds evaluated in the extraction of the antioxidant compounds were ground in a knife mill coupled to a 1 mm diameter circular mesh sieve. A rotational central composite design with eight factorials, six axials and three central points was employed to select the best extraction parameters. The selected independent variables were the concentration of ethanol (Pershy Chemical’s, São Gonçalo, RJ, Brazil) as solvent, temperature, and solid–liquid ratio, according to Table 1. The temperature was limited to 74 °C to avoid loss of solvent. The extraction of antioxidant compounds was performed by agitated solvent extraction, using 125 mL glass flasks properly covered and heated for 1 h under constant stirring of 150 rpm, conditions based on preliminary studies and literature data [15,16]. The experimental data were analyzed by response surface methodology using a second order polynomial equation. Analysis of variance (ANOVA) to test for the lack of fit and coefficient of determination (R^2^) were used to verify the model’s significance.

To determine the best condition for extraction of the antioxidant compounds from *P. cincinnata* defatted seeds, the technique of simultaneous optimization of independent variables (desirability) was used. The desirability function is based on the conversion of each response in an individual desirability (d). After that, they were combined into an overall desirability (D), using the geometric mean. The D value ranges from zero (0) to one (1), in which the value of 1 corresponds to the most desirable response [17]. Under the best operational condition, more assays were performed and observed results were compared with those predicted in order to validate the chosen operational condition.

#### 2.3.1. Total Phenolic Content

Total phenolic content (TPC) was measured according to the method described by Singleton and Rossi [18]. For the assays, 250 µL of each diluted sample was mixed with 1250 µL of 10% Folin–Ciocalteu reagent (Imbralab, Ribeirão Preto, Brazil) for 2 min after 1000 µL of sodium carbonate solution (7.5% *w*/*v*–Dinâmica, Indaiatuba, Brazil) was added. The mixture was incubated for 15 min at 50 °C. Then, the absorbance was measured at 760 nm in a spectrophotometer (5100, Metash, Songjiang District, Shanghai China) vs. a blank prepared with distilled water. Gallic acid (Sigma-Aldrich, Saint Louis, MO, USA) was used as a standard, and the results were expressed as mg gallic acid equivalents per 100 g of sample (mg GAE 100 g^−1^).

#### 2.3.2. Evaluation of the Antioxidant Capacity

ABTS^•+^

The ABTS^•+^ antiradical activity (2,2-azino-bis-(3-ethylbenzthiazoline-6-sulfonic acid-Sigma-Aldrich, Saint Louis, MO, USA) was determined according to methodology reported by Gião et al. [19]. For the reactions, 30 µL of each sample was added to 3 mL of ABTS^•+^. The absorbance was measured at 734 nm in a spectrophotometer (5100, Metash, Songjiang District, Shanghai, China) after 6 min of reaction, using ultra-pure water as blank. The results were expressed as µmol of Trolox g^−1^of sample (Sigma-Aldrich, Buchs, Switzerland).

DPPH^•^

The DPPH^•^ radical (Sigma-Aldrich, Steinheim, Germany) scavenging activity was determined according to Hidalgo et al. [20]. Briefly, 100 µL of each sample was added to 2.9 mL of DPPH^•^ solution (6 × 10^−5^ M in methanol–Vetec, Rio de Janeiro, Brazil) for 30 min. The absorbance was measured in a spectrophotometer (5100, Metash, Songjiang District, Shanghai, China) at 517 nm using methanol as a blank. The DPPH^•^ radical scavenging activity was calculated using Trolox solution (Sigma-Aldrich, Buchs, Switzerland) as a standard, and the results were expressed as µmol of Trolox g^−1^ of sample.

Ferric reducing/antioxidant power (FRAP)

The FRAP assay was performed according to Benzie and Strain [21] with slight modifications. Briefly, three stock solutions were made, 300 mM acetate buffer (pH 3.6), 10 mM TPTZ (Sigma-Aldrich, Buchs, Switzerland) in 40 mM HCl (Isofar, Duque de Caxias, Brazil) and 20 mM FeCl_3_ (Neon, São Paulo, Brazil); for the analysis 25 mL of acetate buffer, 2.5 mL of TPTZ solution, and 2.5 mL of FeCl_3_ were mixed. Then, 100 µL of each sample was reacted with 3 mL of FRAP at 37 °C for 30 min. The absorbance was measured in a spectrophotometer (5100, Metash, Songjiang District, Shanghai, China) at 593 nm using ultra-pure water as a blank. The ferric ion reducing ability was calculated using FeSO_4_⋅7H_2_O (CRQ, Diadema, Brazil) as standard, and the results were expressed as µmol Fe^2+^ g^−1^of sample.

#### 2.3.3. BiocompoundsProfile by UPLC–MS/MS

The defatted seeds extract was analyzed by UPLC–MS/MS using a Nexera UFLC system (Shimadzu) coupled to a high-resolution mass spectrometer with electron spray ionization (ESI), QTOF–MS Impact (Bruker, Billerica, MA, USA). A 1 μL sample (5 mg mL^−1^-HPLC grade methanol, Tedia, São Paulo, Brazil) was injected into an Acquity BEH C18 column (100 mm × 2.1 mm i.d., 1.7 μm particle size) at 40 °C with a constant flow rate of 0.3 mL min^−1^. Mobile phases A (0.1% formic acid in ultrapure water, MilliQ, Merck, Darmstadt, Germany) and B (acetonitrile with 0.1% formic acid) were used in a gradient elution system as follows: the eluent was maintained at 5% B for the first 5 min, followed by a gradient from 5 to 100% B until 39 min, and maintained at 100% B until 44 min. Mass spectra were acquired in positive and negative ion modes, and the analyte was monitored in the *m*/*z* range of 100 to 2000.

The ion source parameters were adjusted under the following conditions: capillary voltage of 4500 V, nebulizer gas pressure of 4.0 bar, desolvation gas flow rate of 10.0 L min^−1^, transfer capillary temperature at 200 °C, and an entrance voltage of −500 V applied to the spectrometer (end plate offset). MS^2^ spectra were acquired in auto MS/MS mode. Data processing was performed using Data Analysis 4.2 software (Bruker). Compound analysis was performed considering the pseudomolecular ion (negative or positive) and mass fragmentation pattern (MS^2^) in manual dereplication and/or comparison with databases (MassBank).

#### 2.3.4. Antimicrobial Assay

In this set of experiments, Gram-negative (*Acinetobacter baumannii* ATCC 19606, *Escherichia coli* ATCC 25922, *Klebsiella pneumoniae* ATCC 13883, and *Pseudomonas aeruginosa* ATCC 27853) and Gram-positive (*Staphylococcus aureus* ATCC 29213, *Staphylococcus epidermidis* ATCC 12228 and *Bacillus subtilis* 168 LMD 74.6) bacteria were grown in Mueller–Hinton agar (Difco, Franklin Lakes, NJ, USA) for 24 h at 35 ± 2 °C. Sabouraud–dextrose agar (Difco, Franklin Lakes, NJ, USA) was used for the cultivation of yeasts, *Candida albicans* ATCC 90028 and *Candida tropicalis* ATCC 750 for 24 h at 35 ± 2 °C. Antimicrobial activity was evaluated using the broth microdilution method with 96-well polystyrene plates, standardized according to documents M07-A9 (for bacterial assays) and M27-A3 (for fungal assays). To determine the minimum bactericidal and fungicidal concentration (MBC and MFC, respectively), 10 μL of the wells that had no visible microbial growth were inoculated in Mueller–Hinton culture medium and Sabouraud-dextrose Agar for 24 h at 37 °C. The MBC and MFC were considered to be the lowest concentration capable of completely inhibiting microbial growth on the agar surface. To eliminate the interference of ethanol in the results, the hydroalcoholic antioxidant extract was subjected to vacuum evaporation. The results were expressed as mg GAE mL^−1^ of ethanol-free extract.

### 2.4. Microencapsulation by Spray Dryer

To evaluate the properties of the microparticles obtained by spray drying, the assays were carried out with three different formulations: (i) oil dispersed in the aqueous phase containing wall material (ii) a mixture of oil and defatted seeds’ ethanol-free antioxidant extract dispersed in the aqueous phase containing wall material and (iii) wall material dispersed in the aqueous phase only.

An oil-in-water emulsion was prepared using 7.5% pressed oil, 7.5% maltodextrin (10DE, MOR-REX 1910, Ingredion, São Paulo, SP, Brazil) and 22.5% modified starch (Capsul^®^06560101CE, Ingredion, São Paulo, SP, Brazil). Modified starch and maltodextrin 10DE were used as wall material in the ratio of 3:1 (*w*/*w*), respectively. The emulsion was homogenized in a blender for 2 min. These operational conditions were adapted from James et al. [22].

The spray dryer was operated under the following conditions: chamber inlet temperature and air velocity of 170 °C and 3 m s^−1^, respectively, emulsion inlet flow rate of 485 mL h^−1^ and 0.7 mm diameter atomization nozzle.

#### 2.4.1. Microparticles’ Oxidative Stability

The oxidative stability of microparticles was measured using a Rancimat^®^ 743 (743, Metrohm, USA), by passing a stream of purified air of 10 L h^−1^ at 110 °C through the oil sample (3 g). Results were expressed as an induction period representing a time interval until the sample reaches a high oxidation level.

#### 2.4.2. X-ray Diffraction

X-ray diffraction (XDR) was performed using a diffractometer (Miniflex, Rigaku, Japan). Samples of the wall material, microencapsulated oil, containing antioxidant extract from defatted seeds or not, were directly analyzed using the equipment. Cu Kα radiation was employed with 2θ ranging from 10° to 80°, with a scan speed of 0.06° 2θ s^−1^.

#### 2.4.3. Morphology of Microparticles

The morphology of microparticles was analyzed by scanning electron microscopy (SEM) (TM303 plus, Hitachi, Japan). The powder samples were mounted on a specimen holder using double scotch tape under vacuum. The microscope was operated at 15 kV.

#### 2.4.4. Fourier Transform Infrared Spectroscopy (FTIR)

The analysis of the chemical structures was performed by FTIR (Nexus 470, Themo Nicolet, USA). The sample powder was blended with spectroscopy-grade KBr (Vetec, Rio de Janeiro, Brazil) and pressed into pellets, and the oil was dripped in a KBr window. The FTIR spectrum of samples was measured from 4000 to 400 cm^−1^, with a resolution of 4 cm^−1^ and 32 accumulations.

#### 2.4.5. Particle Size

A particle size analyzer was used to determine the size distribution of microparticles using laser diffraction (1064, Cilas, France) coupled with ultrasound to increase the dispersibility of the sample. A small amount of powder was suspended in isopropanol and submitted to five particle size distribution readings. The spread of particle sizes was calculated as the scattering according to Equation (2):(2)$$\mathrm{span}=\left(\frac{D_{90}-D_{10}}{D_{50}}\right)$$
where D_10_, D_50_, D_90_ are defined as diameters corresponding to the 10, 50, and 90% cumulative volumes, respectively. The results were expressed in micrometers.

#### 2.4.6. Moisture Content

The moisture content of microparticles was determined by the gravimetric method at 105 °C. The results were expressed as a percentage [23].

#### 2.4.7. Hygroscopicity

Hygroscopicity was determined according to Cai and Corke [24]. Approximately 1 g of sample was placed at 25 ± 2 °C in a desiccator with a saturated NaCl solution (73% relative humidity), and the weight gain due to moisture absorption was measured after one week. The results were expressed in absorbed moisture percentage.

#### 2.4.8. Solubility

The solubility of the microparticles in water was evaluated according to Cano-Chauca et al. [25]. One gram of sample was added to a beaker containing 100 mL distilled water; which was stirred at high speed for 5 min, followed by centrifugation at 3000× *g* for 15 min. After that, a 25 mL aliquot of the supernatant was dried at 105 °C until constant weight. The solubility was calculated by the weight difference and expressed as a percentage.

### 2.5. Statistical Analysis

All analytical determinations were carried out at least in triplicate, except the RANCIMAT and the UPLC-MS/MS trials. Analysis of variance (ANOVA) followed by Fisher’s LSD test was performed using Statistica^®^ v.13.0.

## 3. Results and Discussion

### 3.1. Oil Extraction and Characterization

#### 3.1.1. Yield

The oil content obtained in a lipid extractor using petroleum ether as solvent was about 15%. This result was used as the basis for calculating the efficiency of the process. Lopes et al. [26] found values between 16.7 and 19.2% for lipid content of *P. cincinnata* seeds. The trials were performed with fruits from different accessions, and the oil extraction was conducted in a lipid extractor after moisture removal. The lipid content in *P. cincinnata* was significantly different compared to other passion fruit species. The value obtained was lower when compared with data reported by De Paula et al. [27] for *P. setacea* (31 to 34% of oil/93% efficiency of pressing) and *P. alata* (23% of oil/84% efficiency of pressing) species. Extraction of *P. cincinnata* seeds oil by pressing showed an efficiency of 79%, using seeds with 11% moisture content. The moisture content was selected from preliminary experiments and literature data [28].

#### 3.1.2. Fatty Acid Profile

The fatty acid profile of *P. cincinnata* seeds oil is shown in Table 2. The seeds oil is rich in unsaturated fatty acids, represented mainly by linoleic acid. The observed profile is similar to those reported in the literature by Araujo et al. [5] and Lopes et al. [26]. Linoleic acid is an essential fatty acid since human metabolism does not synthesize it. Therefore, it must be obtained through food intake. It is important to stress that the consumption of polyunsaturated fatty acids may decrease the risk of developing cardiovascular diseases [29]. However, due to the unsaturation in their structure, polyunsaturated fatty acids are very susceptible to oxidation.

### 3.2. Antioxidant Compound Recovery from Defatted Seeds

#### 3.2.1. Evaluation of the Extraction Process

The TPC content and antioxidant capacity of extracts from *P. cincinnata* defatted seeds are shown in Table 1. Trial 10 performed at 74 °C, 55% ethanol as solvent, and 1:35 g mL^−1^ solid–liquid ratio, presented the highest values for TPC, ABTS^•+^, DPPH^•^, and FRAP (2538 mg GAE 100 g^−1^ of sample, 178 µmol Trolox g^−1^ of sample, 370 µmol Trolox g^−1^ of sample, and 795 µmol Fe^2+^ g^−1^ of sample, respectively). The lowest values were found in trial 12 at 53 °C, 97% ethanol in the same solid–liquid ratio (377 mg GAE 100 g^−1^of sample, 21 µmol Trolox g^−1^ of sample, 41 µmol Trolox g^−1^ of sample, 103 µmol Fe^2+^ g^−1^ of sample, respectively). Thus, the results obtained in trial 10 were 6.7, 8.5, 9 and 7.7 times higher than those observed in trial 12, showing that the responses were strongly influenced by temperature and the percentage of ethanol employed in the extraction. Additionally, as can be seen in Appendix A, the results of the experimental design showed a positive correlation (*p* < 0.05), indicating that the increase in the concentration of total phenolic compounds produces an extract with higher antioxidant capacity as measured by different methods.

The temperature was the most relevant parameter, as shown in the Pareto diagram (Figure 1). Its linear effect was significant and positive. In this way, higher temperatures favor the solubility of phenolic compounds and decrease the viscosity of the extraction system, leading to a better extraction yield [16,30]. Notably, this behavior was also observed for the antioxidant capacity of the extracts regardless of the method used (Table 1, Appendix A).

Leal et al. [31] obtained extract of *P. cincinnata* seeds with a total phenolic content of about 2528 mg mg GAE 100 g^−1^, when the assays were conducted at room temperature, 95% ethanol, for 72 h. In the current study, the value found at 74°C, 55% ethanol, solid–liquid ratio of 1:35 for 1 h was similar (2538 mg mg GAE 100 g^−1^ 100 g^−1^ of sample). However, the selected operational conditions present as advantages a shorter extraction time and a lower ethanol concentration in the extraction solution.

Table 1 shows that extraction solutions with intermediate polarity (between 55 to 80% ethanol) were more suitable for extracting antioxidant compounds from defatted seeds. This occurs as a function of the polarity of the recovered compounds. The quadratic effect of this factor was significant (Figure 1), corroborating that there is a maximum value of ethanol concentration which promotes higher attainment of phenolic compounds and improves the antioxidant capacity of the extracts, which can also be seen in the Pareto diagrams for ABTS^•+^, DPPH^•^ and FRAP assays (Appendix A). Shi et al. [32], when extracting phenolic compounds from grape seeds, reported that the best ethanol concentration in the extraction solution was between 55% and 65%. Alcântara et al. [33] evaluated the extraction of phenolic compounds in chia seeds using water, ethanol and acetone in different proportions. In this study, the authors observed that the polarity of the extraction solution significantly influenced the recovery of phenolic compounds. The best solvent mixture was prepared with 17% water, 17% ethanol and 66% acetone.

The solid–liquid ratio was significant for TPC and FRAP responses (Figure 1 and Appendix A). However, this factor made a low contribution to the results. From Table 1, it is possible to verify that results obtained by varying only the solid–liquid ratio, as in the trials 1 and 2, 3 and 4, 5 and 6, 7 and 8 and 13 and 14, the increment in the total phenolic compounds concentration ranged from 1.1 to 1.3 times only (trials 1 and 2; 5 and 6).

All models were significant for predicting the behavior of the responses in relation to the independent variables, as the calculated F-values were higher than the listed F-values (F_9,7_ = 3.68) at *p* = 0.05. The calculated F-values for TPC, ABTS^•+^, DPPH^•^ and FRAP responses were 10.2, 11.3, 8.5, and 11.8, respectively. However, some lack of fit was observed for DPPH^•^ and FRAP responses as it showed *p*-value < 0.05, and the calculated F-values were lower than the listed F-value for this parameter. The R^2^ values of the fitted models were 0.93, 0.94, 0.91, 0.94 for TPC, ABTS^•+^, DPPH^•^ and FRAP responses, respectively, showing that the models accounted for at least 91% of data variability obtained by this experimental design. The R^2^ of the adjusted models was superior to 0.81, reinforcing the good fit of the data, although the DPPH^•^ and FRAP responses have shown some lack of fit. As can be seen in Appendix A, which describe the values observed and those predicted, there is a good agreement between them, making it possible to use these responses to choose the best operational condition to recover antioxidant compounds from the residue.

The desirability tool was employed to obtain the best operational condition for recovering antioxidant compounds from defatted seeds. In this case, the overall desirability was 0.9985 (Figure 2); the closer to 1, the more statistically reliable the result. It indicated as the best operational condition 74 °C, 58% hydroethanolic solution and a solid–liquid ratio of 1:54. However, in order to save solvent in the process, we selected a solid–liquid ratio lower than that estimated by the statistical tool (1:54), since a solid–liquid ratio higher than 1:48 does not significantly increase the recovery of antioxidant compounds from residue. In the interval of the present work (1:48–1:60/Figure 2), there was no significant increase in the recovery of antioxidant compounds from defatted seeds that would justify the increase in solvent in the process. Therefore, the solid–liquid ratio of 1:48 was adopted in the current study. Thus, the antioxidant compound-rich extract constitutes another bioproduct obtained from the *P. cincinnata* seeds.

In this operational condition, observed values of TPC (2868 mg GAE 100 g^−1^ of sample) and antioxidant capacity by ABTS^•+^ (195 µmol Trolox g^−1^ of sample), DPPH^•^ (368 µmol Trolox g^−1^ of sample), and FRAP assays (694 µmol Fe^2+^ g^−1^ of sample) were close to the predicted values by the models as follows: TPC (2952 mg GAE 100 g^−1^ of sample) and antioxidant capacity by ABTS^•+^ (177 µmol Trolox g^−1^ of sample), DPPH^•^ (383 µmol Trolox g^−1^ of sample) and FRAP assays (795 µmol Fe^2+^ g^−1^ of sample) with coefficients of variation less than 10%. The dataset reinforces the operational condition selected for the recovery of antioxidant compounds from *P. cincinnata* defatted seeds.

#### 3.2.2. Biocompound Profile

The chemical composition of the crude extract from *P. cincinnata* seeds was analyzed by UPLC–MS/MS (Table 3). The compound identification was performed by manual dereplication, allowing the putative annotation of eight new lignans in the genus *Passiflora* (**1**–**8**). The base peak chromatographic profile in negative ionization mode can be seen in Figure 3. The MS/MS fragmentation spectra of the major compounds showed the main product ions *m*/*z* 165, *m*/*z* 147 and *m*/*z* 135, suggesting the presence of lignans analogous to secoisolariciresinol and berchemol [34,35]. These two compounds have already been described in the genus *Passiflora* [36,37]. The MS/MS data, combined with the exact mass data of the pseudomolecular ions [M − H]^−^ suggested 3-demethoxy derived compounds.

The main fragment *m*/*z* 165 (C_9_H_9_O_3_^−^) results from the cleavage of the C8-C8′-carbon, followed by loss of CH_4_. The two major lignans (*m*/*z* 327.1280 and *m*/*z* 329.1437) were identified as tetrahydrofuranolignans (**2** and **3**). The other six lignans comprise glycosylated structures, including one tetrahydrofurano lignan *(***1**) and five butanediol lignans (**4**–**8**). In addition to lignans, two glycosylated flavonoids were identified in positive ionization mode. Comparison with MS/MS fragmentation databases allowed the annotation of isovitexin 2″-O-arabinoside (**9**, *m*/*z* 565.1555, [M + H]^+^) and adonivernith (**10**, *m/z* 581.1507, [M + H]^+^) (Figure 4). The elimination of 132 e 252 u revealed the pentose and hexose moiety; however, the glycan stereochemistry could not be determined by LC–MS. Different glycosylated flavonoids have already been reported for *P. cincinnata* [31]. All the annotated compounds have phenolic hydroxyl groups which may contribute to the antioxidant capacity of the extract [38].

#### 3.2.3. Antimicrobial Action

The antimicrobial activity of the *P. cincinnata* defatted seeds extract (obtained in better conditions) was tested against various microorganisms, including yeasts and both Gram-positive and Gram-negative bacteria (Table 4). Initially, we performed the MIC assay. However, the extract interfered with the reading due to its high turbidity. Based on this, we decided to plate 10 µL of each system (untreated and treated with different concentrations of the extract) onto the surface of solid media to observe the bactericidal (MBC)/fungicidal (MFC) effects. The results showed that the *P. cincinnata* defatted seeds extract exhibited bactericidal effects against all tested Gram-positive bacteria (*S. aureus, S. epidermidis* and *B. subtilis*), with MBC values ranging from 0.302 to 0.602 mg GAE mL^−1^ (Table 4). The extract also inhibited the growth of Gram-negative bacteria, except for *P. aeruginosa*, with an MBC value of 0.602 mg GAE mL^−1^, while the growth of fungal cells was not affected by the extract under the employed experimental conditions.

Several studies have attributed the inhibitory effect of seed extracts against different bacteria to their phenolic compounds [15]. These compounds have the ability to bind with the bacterial cell wall and then inhibit bacterial growth. Additionally, phenolic compounds may precipitate proteins and inhibit the enzymes of microorganisms. Siebra et al. [3] studied the effect of a hydroethanolic extract of different parts of *P. cincinnata* Mast against *S. aureus* and *E. coli* and did not observe antimicrobial activity. However, the authors combined the extract of each part with an antimicrobial to reduce the resistance of these microorganisms to the antibiotics, and the results were successful for this type of application. The antibacterial activity observed in the current work may be related to the phytochemical profile of the extract, as revealed by LC–MS/MS (lignan-rich) (Table 3) [39].

### 3.3. Microparticle Characterization

#### 3.3.1. Oxidative Stability of Oil Microparticles

The induction times of *P. cincinnata* pure oil and microencapsulated *P. cincinnata* oil with and without antioxidant extract are shown in Table 5. An oxidative stability of 5.37 h for pure oil was found. This value falls within the range reported by Reis et al. [7], who evaluated the oxidative stability of *Passiflora* species seeds oils recovered by continuous pressing, including *P. alata*, *P. tenuifila*, and *P. setacea* (3.5–7.3 h). These values are typical for passion fruit seeds oils, which are rich in polyunsaturated fatty acids.

It can also be observed that pure oil had oxidative stability increased by about 30% when it was microencapsulated after addition of the antioxidant extract. These results confirm the positive effects of bioactive compounds contained in defatted seeds on the oxidative stability of the microparticles, acting together with the wall material to protect the nutritional and chemical quality of the *P. cincinnata* seeds oil. It is essential to highlight that oil microparticles, with added (or not) antioxidant compounds from defatted seeds, are bioproducts of interest for the food and pharmaceutical industries. Therefore, their preparation can add value to the Brazilian passion fruit agro chain.

#### 3.3.2. DRX

Figure 5 shows the diffractograms obtained for the wall material and *P. cincinnata* oil microencapsulated with and without antioxidant extract.

An amorphous profile was observed for all samples, with non-crystalline character, with only one characteristic signal near 20°. This verifies that incorporating *P. cincinnata* seeds oil and antioxidant extract in the microparticles did not influence the amorphous character typical of polysaccharides [40]. According to Pereira et al. [40], amorphous systems are better for microencapsulation, as they can form a glassy structure by removing water. Additionally, these systems dissolve more easily than those that contain crystalline components since the dissolution of the crystals occurs only on the surface exposed to the solvent. Microparticles of fish oil encapsulated with inulin, isolated whey protein, and maltodextrin by Botrel et al. [41] also exhibited an amorphous structure with a minimum of organization, based on the occurrence of significant diffuse peaks near 20°.

#### 3.3.3. Morphology and Particle Size

As shown in Figure 6, most microparticles were spherical without cracks or fissures, ensuring better protection of the bioactive compounds (oil and phenolic compounds). Particle size distribution at different feed compositions is shown in Table 6. In general, adding antioxidant extract to feed emulsions resulted in the largest particle size (*p* < 0.05), although the distribution was more homogeneous (lowest span values, *p* < 0.05). The microparticles formed by emulsions containing oil and wall material and those containing only wall material were smaller and more heterogeneous than microparticles from emulsions containing oil, antioxidant compounds and wall material. This can be explained as a function of the emulsion’s viscosity. More viscous emulsions give rise to larger droplets, favoring obtaining larger microparticles. This pattern was also reported by Tonon et al. [42], when evaluating the effect of air temperature and feed composition on the particle size distribution of flaxseed oil microparticles.

#### 3.3.4. FTIR

The absorption spectra from FTIR analysis of pure *P. cincinnata* seeds oil, oil microparticles with and without antioxidant extract and wall material are shown in Figure 7. The *P. cincinnata* seeds oil spectra revealed a strong vibrational mode associated with 3009 cm^−1^ which referred to C-H sp^2^ stretching, and intense bands at 2854 and 2925 cm^−1^ which were attributed to the symmetric and asymmetric axial deformation (stretching) of C-H bonds of the methyl (CH_3_) and methylene (CH_2_) of the fatty acid in triacylglycerol. The band at 1747 cm^−1^ corresponds to vibrations of stretching of the C=O group. The band near 1163 cm^−1^ corresponds to the C-O ester group. Bands at 1667–1640 cm^−1^ can be assigned to overlapping of the olefinic C=C stretching and O-H (water). The wall material presents a prominent band at 3384 cm^−1^ related to stretching O-H. The phenolic compounds were in small quantities, so they could not be identified in the analysis, which may be due to the overlapping of their characteristic bands common to other substances in the formulation. However, they were confirmed by LC–MS analysis (Table 3). The absorption spectra for microencapsulated samples, the broad band between 3650 and 3100 cm^−1^ was observed and refers to the wall material used. The characteristic signals of the *P. cincinnata* seeds oil was also observed, characterizing its incorporation throughout the structure of the microparticles. The signals observed were the stretching band of saturated alkanes near 2925 cm^−1^, and carbonyls groups near 1747 cm^−1^, confirming the presence of fatty acids esters. The absorption band at 720 cm^−1^ is attributed to the symmetric stretching vibration of (CH_2_)_n_ groups of n greater than four, indicating the presence of long hydrocarbon linear chains from *P. cincinnata* seeds oil. Thus, the incorporation of the oil into the microparticles can be inferred, given its presence being confirmed by the characteristic peaks of the oil in the absorption spectrum of the microparticles [43,44].

#### 3.3.5. Moisture, Solubility and Hygroscopicity

The moisture, solubility and hygroscopicity of *P. cincinnata* seeds oil microparticles are presented in Table 6. The moisture values ranged from 4.10% to 6.16%. Moisture data are essential since high moisture could favor microparticle oxidation and reduce their stability. Hijo et al. [45] reported that for dry products such as microparticles used in the food industry, the ideal moisture range is between 3 and 4%, where deterioration by microorganisms is reduced. Drying operational conditions, soluble solid content, and oil concentration influence microparticles’ moisture, which explains the data observed in the present work. In this way, the mixture containing wall material and water forms a more hygroscopic solution when compared to oil/water/wall material emulsions with or without the addition of the antioxidant extract. Thus, the probability of the microparticles containing only the wall material, retaining more water during the process is higher, as they tend to come into equilibrium with the ambient humidity more easily.

The hygroscopicity values of the microparticles ranges from 7.17 to 10.75% (Table 6). As oil and water are immiscible, the oil microparticles presented smaller hygroscopicity values than those from wall material–water mixture. According to Nurhadi et al. [46], microparticles with hygroscopicity higher than 20% can be considered very hygroscopic; therefore, they are not stable, being more susceptible to water absorption and, to the stickiness of the material and oxidation of the target compounds [13].

The solubility was higher than 76% for oil microparticles with or without antioxidant compounds (Table 6). The addition of antioxidant extract did not affect the solubility of samples (*p* < 0.05). These results are according to de Oliveira et al. [15], who elaborated microparticles of buriti oil by freeze–drying using carbohydrates as wall material, whose highest value reported was 71%. Botrel et al. [41] reported 79% solubility for microparticles of fish oil atomized in spray dried using different wall materials, where the microparticles were found to have good solubility. In this way, *P. cincinnata* seeds oil microparticles demonstrate good solubility in water and, therefore, have potential for application in aqueous systems.

## 4. Conclusions

This study was successful in obtaining *P. cincinnata* seeds oil by pressing and recovering antioxidant compounds from the defatted seeds. The best operational conditionsto obtain an antioxidant extract of defatted seeds was 74 °C, 58% ethanol as solvent, and a solid–liquid ratio of 1:48. The main compounds identified by UPLC MS–MS in the extract were lignans that may contribute to antioxidant and antimicrobial activities. The microencapsulation was adequate to preserve *P. cincinnata* oil, and the addition of antioxidant extract proved to be a great method to increase the oxidative stability. Thus, the present work may contribute to adding value to the Brazilian Caatinga passion fruit agro chain by obtaining three bioproducts: pure oil, antioxidant extract and oil microparticles with antioxidant extract.

## Figures and Tables

**Figure 1 foods-12-02525-f001:**
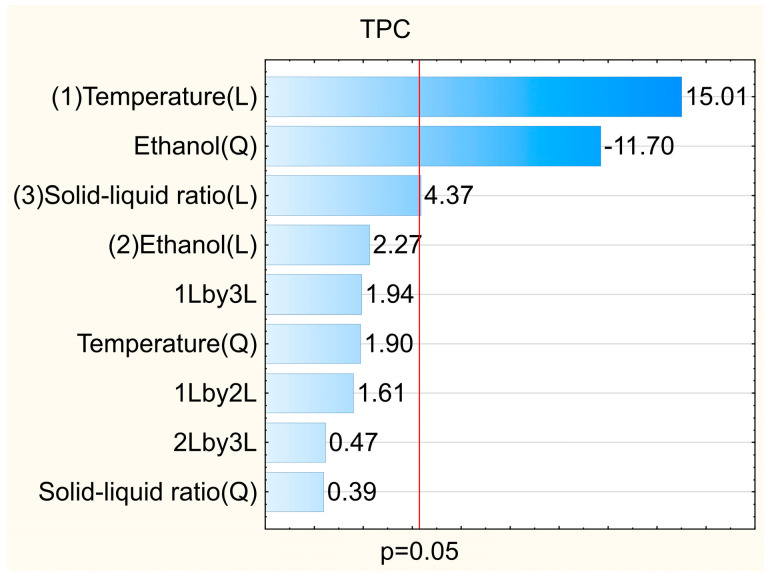
Influence of the independent variables on the total phenolic compounds in the extract of *Passiflora cincinnata* defatted seeds.

**Figure 2 foods-12-02525-f002:**
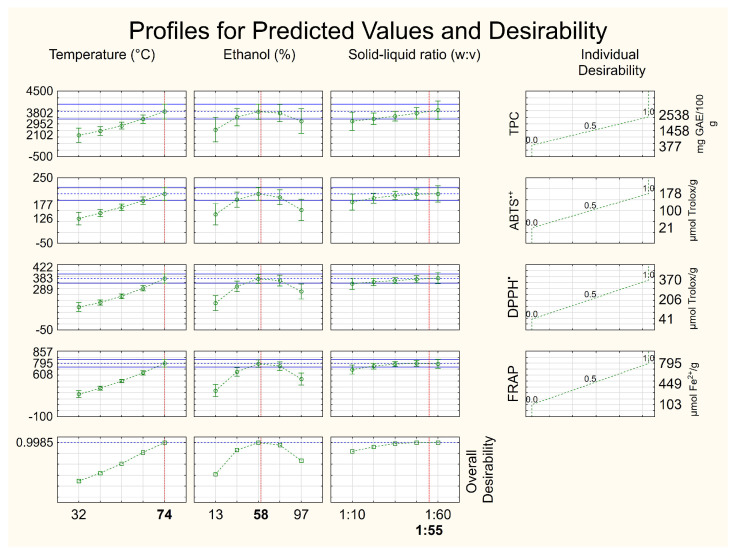
Profiles for predicted values and overall desirability for independent variables temperature, ethanol concentration and solid–liquid ratio employed in the extraction of antioxidant compounds from *Passiflora cincinnata* defatted seeds.

**Figure 3 foods-12-02525-f003:**
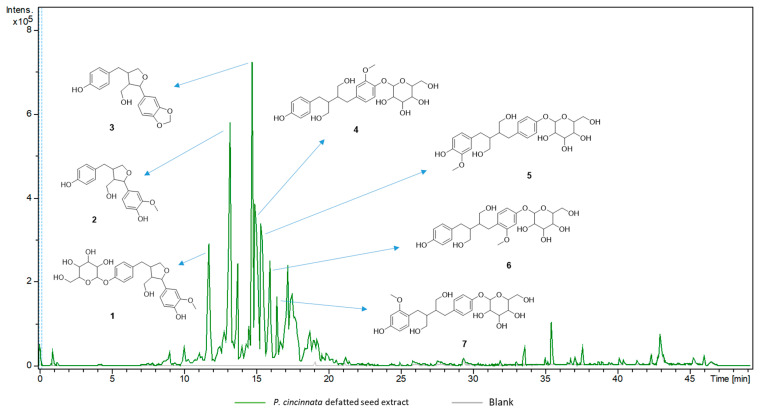
Base peak chromatogram (UPLC–MS/MS) in negative ionization mode of *Passiflora cincinnata* defatted seeds extract and the respective lignans (**1**–**7**) annotated for base peaks. The diglycosylated lignan **8** (Figure 4) is overlapped with the major ion signal at *m*/*z* 327.1280.

**Figure 4 foods-12-02525-f004:**
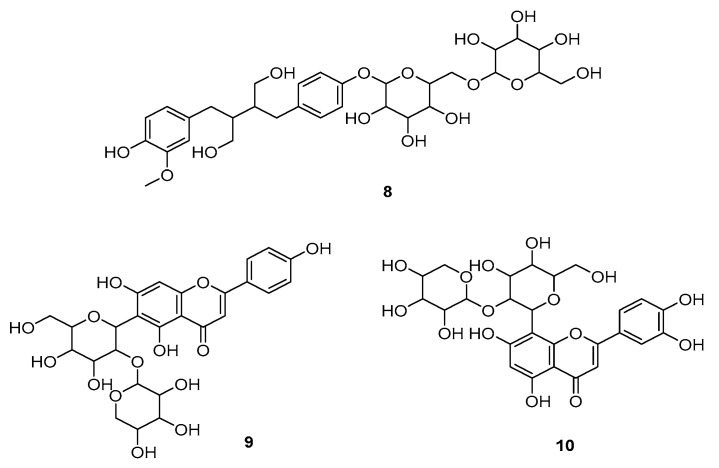
Minor compounds annotated in the *Passiflora cincinnata* defatted seeds extract.

**Figure 5 foods-12-02525-f005:**
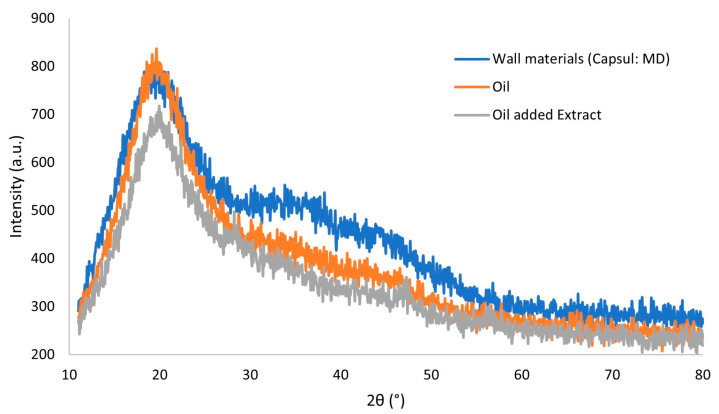
Diffractogram of the microparticles containing *Passiflora cincinnata* seeds oil (orange); microparticles containing *Passiflora cincinnata* seeds oil and antioxidant extract (gray); microparticles of wall material only (blue).

**Figure 6 foods-12-02525-f006:**
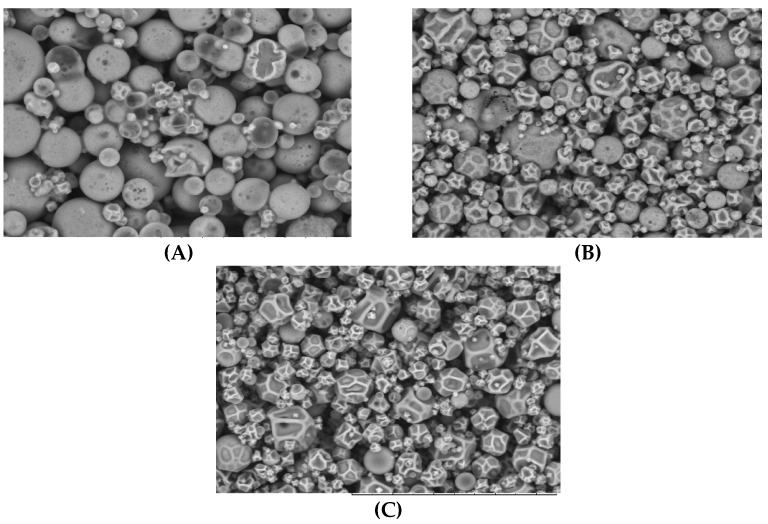
Micrograph of microparticles containing *Passiflora *cincinnata** seeds oil (**A**); microparticles containing *Passiflora *cincinnata** seeds oil plus antioxidant extract (**B**); microparticles of wall material only (**C**), visualized by MEV.

**Figure 7 foods-12-02525-f007:**
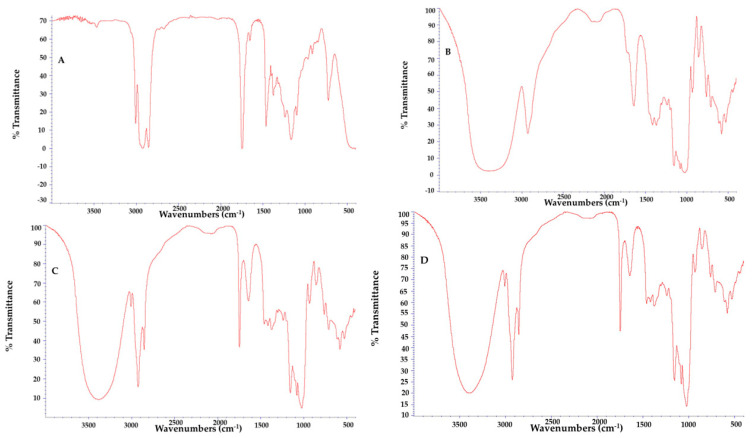
FTIR spectra of *Passiflora cincinnata* seeds oil (**A**) wall material (**B**) microparticles containing *Passiflora cincinnata* seeds oil and antioxidant extract (**C**) microparticles containing *Passiflora cincinnata* seeds oil (**D**).

**Table 1 foods-12-02525-t001:** Total phenolic compounds content and antioxidant capacity of the hydroethanolic extract of *Passiflora cincinnata* defatted seeds obtained under different experimental conditions.

Trials	Temperature (°C)	Ethanol (%)	Solid/Liquid Ratio (g mL^−1^)	TPC ^1^	ABTS^•+ 2^	DPPH^• 2^	FRAP ^3^
1	40 (−1)	30 (−1)	1:20 (−1)	845 ± 25	43 ± 2	111 ± 3	223 ± 8
2	40 (−1)	30 (−1)	1:50 (+1)	912 ± 31	51 ± 1	107 ± 5	216 ± 4
3	40 (−1)	80 (+1)	1:20 (−1)	1016 ± 27	64 ± 3	148 ± 6	284 ± 4
4	40 (−1)	80 (+1)	1:50 (+1)	1220 ± 25	69 ± 1	153 ± 6	304 ± 6
5	65 (+1)	30 (−1)	1:20 (−1)	1371 ± 57	80 ± 2	182 ± 8	395 ± 8
6	65 (+1)	30 (−1)	1:50 (+1)	1799 ± 119	105 ± 2	214 ± 12	462 ± 19
7	65 (+1)	80 (+1)	1:20 (−1)	1870 ± 35	114 ± 2	280 ± 11	567 ± 18
8	65 (+1)	80 (+1)	1:50 (+1)	2302 ± 31	130 ± 2	292 ± 10	614 ± 17
9	32 (−1.68)	55 (0)	1:35 (0)	991 ± 42	63 ± 4	128 ± 7	303 ± 6
10	74 (+1.68)	55 (0)	1:35 (0)	2538 ± 141	178 ± 6	370 ± 20	795 ± 5
11	53 (0)	13 (−1.68)	1:35 (0)	720 ± 18	45 ± 1	92 ± 5	211 ± 2
12	53 (0)	97 (+1.68)	1:35 (0)	377 ± 17	21 ± 1	41 ± 0	103 ± 1
13	53 (0)	55 (0)	1:10 (−1.68)	1451 ± 69	99 ± 4	210 ± 5	474 ± 17
14	53 (0)	55 (0)	1:60 (+1.68)	1809 ± 80	111 ± 8	226 ± 5	462 ± 16
15	53 (0)	55 (0)	1:35 (0)	1810 ± 124	121 ± 1	235 ± 1	515 ± 6
16	53 (0)	55 (0)	1:35 (0)	1616 ± 57	117 ± 1	222 ± 4	527 ± 1
17	53 (0)	55 (0)	1:35 (0)	1637 ± 64	110 ± 3	234 ± 3	504 ± 3

The coded values of the independent variables are in parentheses. Results expressed as mean ± standard deviation (coefficient of variation < 10%). TPC—Total phenolic compounds; ^1^ Results expressed as mg GAE 100 g^−1^ of sample; ^2^ Results expressed as µmol Troloxg^−1^ of sample; ^3^ Results expressed as µmol Fe^2+^ g^−1^ of sample.

**Table 2 foods-12-02525-t002:** Fatty acid profile of *Passiflora cincinnata* seeds oil expressed as percentage (%).

Fatty Acids	Present Work	Lopes et al. [25]	Araújo et al. [5]
Palmitic (C16:0)	12.14 ± 1.00	10.2	9.2
Stearic (C 18:0)	1.09 ± 0.26	2.9	3.0
Oleic (C 18:1)	8.43 ± 1.32	11.3	15.4
Linoleic (C 18:2)	78.34 ± 2.22	74.3	70.3
Linolenic (C 18:3)	-	0.6	0.6

**Table 3 foods-12-02525-t003:** Compounds annotated by UPLC–MS/MS (ESI positive and negative modes) in *Passiflora cincinnata* defatted seeds extract.

Compound Name	T_R_ (min)	Molecular Formula	Adduct Ion	Experimental *m*/*z*	Main MS^2^Fragment Ions	Relative Percentage (%)
3-demethoxy-8-dehydroxy-berchemol 4-*O*-glucoside (**1**)	11.8	C_25_H_32_O_10_	[M − H]^−^	491.2162	165.0786, 147.0655, 135.0650	5.9
3-demethoxy-8-dehydroxy-berchemol (**2**)	13.2	C_19_H_22_O_5_	[M − H]^−^	329.1431	165.0793, 147.0660, 146.0572, 135.0659, 129.0540	10.7
2-(1,3-benzodioxol-5-yl)tetrahydro-4-[(4-hydroxyphenyl)methyl]-3-furanmethanol (**3**)	14.8	C_19_H_20_O_5_	[M − H]^−^	327.1280	163.0624, 162.0538, 147.0657, 135.0649,	8.5
Secoisolariciresinol 4-*O*-xylopyranoside (**4**)	15.0	C_25_H_34_O_10_	[M − H]^−^	493.2113	329.1427, 299.1287, 179.0591, 165.0777, 147.0656, 137.0443	10.6
3-demethoxy-secoisolariciresinol 4-*O*-glucoside (**5**)	15.5	C_25_H_34_O_10_	[M − H]^−^	493.2108	329.1429, 299.1292, 165.0779, 147.0661, 137.0445	9.8
2-methoxy-bisdemethoxy- secoisolariciresinol 4-*O*-glucoside (**6**)	16.0	C_25_H_34_O_10_	[M − H]^−^	493.2107	165.0781, 147.0658, 135.0650	4.9
2′-methoxy-bisdemethoxy- secoisolariciresinol 4-*O*-glucoside (**7**)	16.5	C_25_H_34_O_10_	[M − H]^−^	493.2102	165.0780, 147.0657, 135.0651	2.6
3-demethoxy-secoisolariciresinol 4-*O*-glucosil glucoside (**8**)	14.8	C_31_H_44_O_15_	[M − H]^−^	655.2615	165.0778, 163.0622, 162.0543	7.2
Isovitexin 2″-*O*-arabinoside (**9**)	11.5	C_26_H_28_O_14_	[M + H]^+^	565.1555	283.0595, 313.0702, 397.0917, 415.1010, 433.1253	<0.5
Adonivernith (**10**)	10.8	C_26_H_28_O_15_	[M + H]^+^	581.1507	299.0564, 329.0659, 413.0844, 431.1016, 449.1073	<0.5

**Table 4 foods-12-02525-t004:** Antimicrobial activity of *Passiflora cincinnata* defatted seeds extract.

Microorganisms	MBC/MFC [mg GAE mL^−1^] ^a^
Gram-positive bacteria	
*Bacillus subtilis* 168 LMD 74.6	0.602
*Staphylococcus aureus* ATCC 29213	0.302
*Staphylococcus epidermidis* ATCC 12228	0.302
Gram-negative bacteria	
*Acinetobacter baumannii* ATCC 19606	0.602
*Escherichia coli* ATCC 25922	0.602
*Klebsiella pneumoniae* ATCC13883	0.602
*Psedomonas aeruginosa* ATCC 27853	ND
Fungi	
*Candida albicans* ATCC 90028	ND
*Candida tropicalis* ATCC 750	ND

ND—not determined. ^a^ Results expressed as mg gallic acid equivalent/mL of ethanol-free extract. MBC—minimum bactericidal concentration. MFC—minimal fungicidal concentration.

**Table 5 foods-12-02525-t005:** Induction time of *P. cincinnata* seeds oil and its microparticles.

Samples	Induction Time (Hours)
Pure oil	5.37± 0.18 ^b^
Oil + antioxidant extract microencapsulated	6.97± 0.73 ^a^
Oil microencapsulated	5.27± 0.53 ^b^

Different letters indicate significant differences.

**Table 6 foods-12-02525-t006:** Particle size parameters and moisture, hygroscopicity and solubility values of *Passiflora cincinnata* seeds oil microparticles.

Samples	Average Particle Size (μm)	Span Value	Moisture (%)	Hygroscopicity (%)	Solubility (%)
Oil, phenolic extract and wall material microencapsulated	20.63 ±0.81 ^b^	1.46 ±0.11 ^b^	4.83 ±0.13 ^b^	7.17 ± 0.10 ^b^	76.97 ±0.21 ^a^
Oil and wall material microencapsulated	16.40 ±0.54 ^a^	2.37 ±0.16 ^a^	4.10 ±0.06 ^c^	7.53 ±0.04 ^b^	77.03 ±0.36 ^a^
Wall material microencapsulated	15.50 ±0.05 ^a^	2.22 ±0.06 ^a^	6.16 ±0.12 ^a^	10.75 ±0.77 ^a^	72.87 ±1.74 ^b^

Different letters in the same column indicate significant differences.

## Data Availability

The data used to support the findings of this study can be made available by the corresponding author upon request.

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
