# Peer review of "Bioproducts from Passiflora cincinnata Seeds: The Brazilian Caatinga Passion Fruit"

_foods, 2023, doi:10.3390/foods12132525_

Round 1
Reviewer 1 Report
the manuscript is quite interesting for readers. several comments and suggestions are addressed to improve the quality of manuscript.
1. why do you choose central rotation experimental design instead of box bhenken design? in my opinion. BBD is suitable due to 3 parameters has been used.
2. please paraphrase line 89 to 91
3. stated the chemical used/purchased in the section material and methods
4. the extraction time was 24 hours? how do you know there is no degradation during the extraction?
5.add the subsection optimization in the material and methods
6. add the validation for multiple optimization
Author Response
Reviewer 1
the manuscript is quite interesting for readers. several comments and suggestions are addressed to improve the quality of manuscript.
Answer: Dear reviewer, thanks for your positive and constructive comments. We have addressed the comments and revised the manuscript accordingly. More information and details were included in the text (highlighted in yellow).
- why do you choose central rotation experimental design instead of box bhenken design? in my opinion. BBD is suitable due to 3 parameters has been used.
Answer: Both types of experimental design allow using 3 independent variables. We opted for the central rotational experimental design, as it is composed of 5 levels for each parameter, expanding the study region, by adding axial points to the factorial points from the design. Box Behnken design is also a good alternative. However, it only allows us to evaluate 3 levels for each independent variable. For variables such as ethanol concentration, which has an impact on the polarity of the extraction medium, using more levels makes it easier to understand the results obtained.
- please paraphrase lines 89 to 91
Answer: It is done. Please, see Page 3 and lines 98-101.
- stated the chemical used/purchased in the section material and methods
Answer: Dear reviewer, we have added details of the chemicals used in the work throughout the text. The information includes manufacturer, city and country, in according to the Foods instructions.
- the extraction time was 24 hours? how do you know there is no degradation during the extraction?
Answer: Dear reviewer, the extraction only occurred for an hour. This time was adopted based on Lab’s works and literature (References 15 and 16). We have highlighted this in the text. Please, see item 2.3, pages 3-4.
5.add the subsection optimization in the material and methods
Answer: It is done. Please, see item 2.3, Page 3-4.
- add the validation for multiple optimization
Answer: It is done. Please, see page 9 and lines 399-406.
Reviewer 2 Report
The manuscript is poorly written and difficult to read and understand fluently. In addition, the originality, scientific impact and interest for an expert audience is rather low. Many key parts are absent in the manuscript, and the conclusions are mainly limited to a comparison with the literature. I also find it difficult to exploit the results obtained, particularly for the antioxidant fraction for the reasons described below. The authors should consider the reviewers' opinions, rewrite the manuscript also with the help of a native speaker, and proceed with a new submission.
Table 1 is an inappropriate position in the text. Move to results section.
Table 1. Error deviation and statistical analysis are missing. Explain the means of brackets.
The authors should perform a correlation analysis between TPC and antioxidant activity, explaining the existence or the absence of a correlation
Table 2. Data in the table and figure should not include values from previous work. Analogies or discrepancies should be indicated and explained in the text. Error deviation and statistical analysis are missing.
Discussions of results are almost absent and, when present, are merely referend to previous studies. The authors should better discuss them, emphasizing the effects of experimental conditions on the results obtained.
One major issue concerning the extraction of bioactive compounds is the presence of organic solvents harmful for the human body and the environment. This is one main reason for their gradual substitution with safer compounds such as nades. In this study, the authors used organic extraction with high solid liquid ratio performed at high temperatures. This procedure is not sustainable, particularly for incoming countries such as Brazil. This represents a strong limitation of the applicability of the results obtained.
Table 3. The authors should include the concentration for each polyphenolic compound
Table 4. Error deviation and statistical analysis are not present
What is the rationale for encapsulating oil for seeds? Is it only to prevent lipid peroxidation? The protective effect of microencapsulation on lipid peroxidation has been studied so far (at least 20 years) and does not possess a high scientific echo.
Needs to be checked by native english
Author Response
Reviewer 2
The manuscript is poorly written and difficult to read and understand fluently. In addition, the originality, scientific impact and interest for an expert audience is rather low. Many key parts are absent in the manuscript, and the conclusions are mainly limited to a comparison with the literature. I also find it difficult to exploit the results obtained, particularly for the antioxidant fraction for the reasons described below. The authors should consider the reviewers' opinions, rewrite the manuscript also with the help of a native speaker, and proceed with a new submission.
Answer: Dear reviewer, we have carefully read all the points raised and improved the manuscript according to the suggested corrections. As for the originality of the present work and its contribution for the use of agroindustrial residue, in particular the use of Caatinga passion fruit seeds, this is the first report that addressed the maximum use of this residue, which resulted in microparticles containing oil and the antioxidants from the same residue, obtained sequentially. It is important to emphasize that microencapsulation is a technique that is already widespread and evaluated. However, in this work, it was useful to preserve two bioproducts obtained from Caatinga passion fruit seeds at the same time. Also, it should be noted that the use of natural antioxidants increased the oxidative stability of the caatinga passion fruit seeds oil by 30%, which strengthens the proposal of this work.
Table 1 is an inappropriate position in the text. Move to results section.
Answer: Table 1 was moved to item 3.2 and renumbered as table 2. Please, see page 8.
Table 1. Error deviation and statistical analysis are missing. Explain the means of brackets.
Answer: The standard deviation has been added to Table 2. Data in parentheses are coded values of the evaluated parameters in the experimental design. It was added to the footnote from the table. In this part of the work, it was decided not to apply ANOVA and post Hoc test to verify the difference between the results. We focus on knowing the effects of each independent variable on the evaluated responses by means Pareto diagrams, whose analysis has sufficient statistical framework to explain the data. To better substantiate the findings, we have added to the discussion more details about the statistics from experimental design, such as the significance of the models, coefficient of determination, among others.
The authors should perform a correlation analysis between TPC and antioxidant activity, explaining the existence or the absence of a correlation
Answer: We have performed this analysis. The correlation matrix has been attached as supplementary material. In the manuscript we have added the sentence below:
“Also, as can be seen in Table S1, the results of the experimental design showed a positive correlation (p<0.05), indicating that the increase in the concentration of total phenolic compounds produces extract with higher antioxidant capacity measured by different methods.” Item 3.2.1, Page 7, lines 331-334.
Table 2. Data in the table and figure should not include values from previous work. Analogies or discrepancies should be indicated and explained in the text. Error deviation and statistical analysis are missing.
Answer: We have added the standard deviation to the results of this table, but we did not perform ANOVA and post hoc test because it is about the composition of an oil sample, making no sense to statistically evaluate this result. In addition, we kept the literature data in the table because we believe it facilitates the visualization of similarities and differences between Caatinga passion fruit seeds oils reported in the literature.
Discussions of results are almost absent and, when present, are merely referend to previous studies. The authors should better discuss them, emphasizing the effects of experimental conditions on the results obtained.
Answer: The item 3.2.1 was revised and the effects of experimental conditions were added to discussion as recommended for this reviewer. Also, we have added statistical analysis of the experimental design.
One major issue concerning the extraction of bioactive compounds is the presence of organic solvents harmful for the human body and the environment. This is one main reason for their gradual substitution with safer compounds such as nades. In this study, the authors used organic extraction with high solid liquid ratio performed at high temperatures. This procedure is not sustainable, particularly for incoming countries such as Brazil. This represents a strong limitation of the applicability of the results obtained.
Answer: Brazil is the second largest ethanol global producers in according to report from Renewable Fuels Association (https://ethanolrfa.org/markets-and-statistics/annual-ethanol-production). It is mainly derived from sugarcane, and therefore less expensive than other sources used in the world such as corn (used in the US). Thus, ethanol is considered a renewable organic solvent, another advantage over conventional organic solvents of fossil origin. In this way, it is abundant and lower cost product in the country in comparison with NADES. In addition, ethanol is a solvent widely used in pharmaceutical and food formulations such as propolis extract or vanilla essence. Therefore, its use should be encouraged, and evaluating the best ethanol concentration and other bioactive compound extraction parameters is essential to reduce process costs and possible environmental impacts. In our work, it was observed that 58% ethanol represented the best concentration to extract antioxidant compounds from Caatinga passion fruit defatted seeds. Low and high limits from this concentration reduce extraction efficiency. Figure 2 summarizes the operational condition chosen for the extraction process.
.Table 3. The authors should include the concentration for each polyphenolic compound
Answer: The relative percentages of annotated compounds were added in the Table 3 – page 11.
Table 4. Error deviation and statistical analysis are not present
Answer: This table refers to the evaluation of the antimicrobial activity of an extract on different microorganisms. Therefore, it is not applicable to perform ANOVA and post-hoc test on these data. In addition, for this type of evaluation, it is recommended to present only the mean values because it is a test, which by nature, can present fluctuations. Still, it should be noted that the MBC values were different only for two bacteria, being located a dilution below those found for the other evaluated bacteria.
What is the rationale for encapsulating oil for seeds? Is it only to prevent lipid peroxidation? The protective effect of microencapsulation on lipid peroxidation has been studied so far (at least 20 years) and does not possess a high scientific echo.
Answer: Microencapsulation by spray drying is a consolidated technique and of interest to the food industry as it is a quick way to remove water from food and increase the shelf life of highly perishable food products. Associated to dehydration, the use of wall material works as a protective film that reduces the effects of factors extrinsic to food such as light, temperature and oxygen. That said, the novelty of this work resides in obtaining different bioproducts from the same fraction, the Caatinga passion fruit seeds, which is a fruit depulping residue, and combining them to produce microparticles of oil and natural antioxidants. Thus, microencapsulation by spray drying played an important role in the biorefinery of Caatinga passion fruit seeds, by stabilizing, at the same time, the oil and the antioxidant compounds extracted from the defatted seeds, yielding microparticles with 30% higher oxidative stability when compared to those without the addition of antioxidant extract. It should also be noted that microencapsulation renders products that require less storage space when compared to original products, such as extracts and emulsions. Finally, this is the first study that evaluated this approach for Caatinga passion fruit seeds.
Reviewer 3 Report
My comments connected with the publication of the manuscript submitted for review.
1. The purpose of the work should be clarified. The authors not only obtained specific bioproducts, but also examined their specific chemical and biological properties.
2. In what exact units was the antioxidant activity and polyphenol content of the obtained extracts expressed? What does μmol de Trolox g-1 mean? Is it gram dry matter, extract, etc.?
3. Line 294: DPPH radicals should not contain "+".
4. Line 295: In the Materials and Methods section, the authors used the expression mg GAE 100 g-1. Please unify the unit throughout the manuscript.
5. Are the names of compounds 5 and 8 in Table 3 spelled correctly. Please check.
6. Why was the microbiological activity of the tested extracts expressed as "mg GAE mL-1"?
7. In the "References" section, Latin names of plants should be written in italics
Author Response
Reviewer 3
My comments connected with the publication of the manuscript submitted for review.
Answer: Dear reviewer, thanks for your positive and constructive comments. We have addressed the comments and revised the manuscript accordingly. More information and details were included in the text (highlighted in yellow).
- The purpose of the work should be clarified. The authors not only obtained specific bioproducts, but also examined their specific chemical and biological properties.
Answer: We have rewritten the objective of the work as recommended. Please, see Page 3, lines 98-101.
- In what exact units was the antioxidant activity and polyphenol content of the obtained extracts expressed? What does μmol Trolox g-1mean? Is it gram dry matter, extract, etc.?
Answer: The results were expressed in mg GAE 100 g-1 or μmol Trolox g-1 or μmol Fe2+ g-1 “of sample” (defatted seeds used in extraction). We did not dry the extracts to report the results in dry matter, as this could impact on stability of the antioxidant compounds, being one more variable influencing the results. Thus, as different solid-liquid ratios were used, we took into account the mass of defatted seeds and the volume of solvent employed in each extraction to express the results. We've been complementing the units throughout the work.
- Line 294: DPPH radicals should not contain "+".
Answer: So sorry, it was deleted.
- Line 295: In the Materials and Methods section, the authors used the expression mg GAE 100 g-1. Please unify the unit throughout the manuscript.
Answer: Sorry, we have corrected it throughout the manuscript.
- Are the names of compounds 5 and 8 in Table 3 spelled correctly. Please check.
Answer: Sorry, we have corrected it. Please, see table 3.
- Why was the microbiological activity of the tested extracts expressed as "mg GAE mL-1"?
Answer: This set of experiment was based on the concentration in mg GAE/mL of ethanol-free extract. The ethanol was removed under vacuum in order to preserve antioxidant compounds. Unfortunately, it was not possible to work with dried extract. However, this approach was useful to display the antibacterial potential of the sample. We have added this detail to the manuscript. Please, see item 2.3.4, page 5, lines 228-230.
We like to emphasize that similar works have been published using this unit.
Salaheen, C. Nguyen, C. Mui, D. Biswas, Bioactive berry juice byproducts as alternative and natural inhibitors for Salmonella Gallinarum and Salmonella Pullorum, Journal of Applied Poultry Research, Volume 24, Issue 2, 2015, https://doi.org/10.3382/japr/pfv021
Serajus Salaheen, Zajeba Tabashsum, Stefano Gaspard, Anthony Dattilio, Thomas H. Tran, Debabrata Biswas, Reduced Campylobacter jejuni colonization in poultry gut with bioactive phenolics, Food Control, Volume 84, 2018, https://doi.org/10.1016/j.foodcont.2017.07.021
Antimicrobial and cytotoxic activity to human colon adenocarcinoma cell lines (HT-29) potential of olive oil extraction residue. Vanessa Ferreira do Amaral and Angela Cristina Mello dos Santos and Josué Guilherme Lisboa Moura and Juliana de Castilhos and Tanise Gemelli and Jéssica Fernanda Hoffmann and Valmor Ziegler and Cristiano Dietrich Ferreira. Natural Product Research, 2021, 36, 4486 – 4491. DOI:10.1080/14786419.2021.1986708
- In the "References" section, Latin names of plants should be written in italics
Answer: This section has been throughout checked as recommended.
Reviewer 4 Report
The manuscript "Bioproducts from Passiflora cincinnata seeds: The Brazilian Caatinga passion fruit" aims to obtain three bioproducts (pure oil, antioxidant extract and oil microparticles with antioxidant extract.
The study is interesting and well conducted. However, some changes are needed.
Lines 89-91: From what was said in the introduction, these 3 objectives have already been developed by other authors and therefore the novelty of the study cannot be understood. Authors should explain well the novelty with reference to the studies in the bibliography.
Lines 129-131: Please, explain how and why the level ranges for the 3 factors were chosen
Table 1: Please, enter the standard deviations for the 4 response variables. Also enter the predicted values for the 4 variables, which will then be used to build the RSM model.
Paragraph 3.2.1: In the description of the results, the tables of Analysis of variance for the second-order polynomial equation for the four variables are missing. Tables should also show goodness of fit with R2, R2adj, lack of fit and pure error. The explanation of the analysis techniques used (response surface methodology and desirability function) for Antioxidant compounds recovery should be included in the materials and methods.
It is suggested that you review punctuation in text and captions.
Author Response
Reviewer 4
The manuscript "Bioproducts from Passiflora cincinnata seeds: The Brazilian Caatinga passion fruit" aims to obtain three bioproducts (pure oil, antioxidant extract and oil microparticles with antioxidant extract.
The study is interesting and well conducted. However, some changes are needed.
Answer: Dear reviewer, thanks for your positive and constructive comments. We have addressed the comments and revised the manuscript accordingly. More information and details were included in the text (highlighted in yellow).
Lines 89-91: From what was said in the introduction, these 3 objectives have already been developed by other authors and therefore the novelty of the study cannot be understood. Authors should explain well the novelty with reference to the studies in the bibliography.
Answer: We have revised the introduction in order to clarify the originality of the work. Please, see pages 1-3.
Lines 129-131: Please, explain how and why the level ranges for the 3 factors were chosen
Answer: Factor levels were chosen based on preliminary studies and literature data as described in the text (Please, see item 2.3, Pages 3-4). It is important to emphasize that we chose to evaluate a wide range of polarity (ethanol%), since antioxidant compounds, especially phenolic ones, have varied structures, ranging from simple molecules to very complex ones, which impacts on their solubility. Thus, screening a larger range of ethanol percentages favors finding the optimal region. Temperature also has a strong influence on the extraction processes of antioxidant compounds, so, based on previous studies, we chose to vary from room temperature to 74 °C, since pure ethanol has a boiling point equal to 78.4 °C. In order to know the range where the solvent loses its solubilization capacity, we also chose to evaluate a broader range of solid-liquid ratio.
Table 1: Please, enter the standard deviations for the 4 response variables. Also enter the predicted values for the 4 variables, which will then be used to build the RSM model.
Answer: Standard deviations were added in Table 2. As Table 2 contains a lot of information, we chose to report the predicted values as a figure. The observed versus predicted values graphs for 4 responses are attached as supplementary material.
Paragraph 3.2.1: In the description of the results, the tables of Analysis of variance for the second-order polynomial equation for the four variables are missing. Tables should also show goodness of fit with R2, R2adj, lack of fit and pure error. The explanation of the analysis techniques used (response surface methodology and desirability function) for Antioxidant compounds recovery should be included in the materials and methods.
Answer: As the manuscript already contains many tables and figures, the main data extracted from the ANOVA were added directly to the text (see page, line). A description of the desirability tool was added in the material and methods (see page 4, lines 146-157). For that, reference 17 was added.
- Derringer, G.; Suich, R. Simultaneous optimization of several response variables. J. Qual. Technol. 1980, 12, 214–219.
It is suggested that you review punctuation in text and captions.
Answer: We have revised throughout the manuscript.
Round 2
Reviewer 2 Report
The level of English is still decidedly low, and the study even though conducted on a particular type of by-product still has a significant lack of innovation and originality, particularly in the type of extraction. In addition, the lack of statistical analysis that continues to be avoided by the authors makes it impossible to define whether the variations between experimental conditions are the result of chance or of the condition itself. It is not appropriate to include in tables that normally refer to the results obtained in the study with values extrapolated from other articles. Finally, polyphenol content should be included as concentration and not as relative abundance (among other things, without standard deviation). Finally, the rationale for the use of organic solvents as extractive methods in the face of a unanimous direction from the scientific community toward more environmentally friendly solvents is insufficient.
In conclusion, this paper should be reworded and rewritten in its entirety and resubmitted
Extensive editing of English language required
Reviewer 4 Report
The authors significantly improved the manuscript